# Effect of Olivine Additive on the Shear Resistance of Fine-Grained Soils: A Sustainable Approach for Risk Mitigation and Environmental Impact Reduction

Abdelmaoula Mahamoud Tahir 🆔 and Sedat Sert *🆔

Department of Civil Engineering, Sakarya University, Sakarya 54050, Turkey;
abdelmaoula.tahir@ogr.sakarya.edu.tr
* Correspondence: sert@sakarya.edu.tr

**Abstract:** Increasing urbanization has also accelerated the potential risks of hazards arising from problematic soils. At this end, it becomes inevitable to apply soil improvement methods, which are the most applicable and economical methods. Increasing the strength of clays, especially in undrained conditions where they exhibit low shear resistance, is essential for mitigating soil-induced hazards. This study aimed to improve the shear resistance of clays with a waste material named olivine, which has rarely been used in soils before. The undrained shear strength of the samples prepared at different curing times was determined at various confining pressures. Both olivine-added and potassium hydroxide (KOH)-activated olivine-added samples were tested in the same manner. It was proven that the olivine contribution alone was not sufficient over time, and higher shear resistance was obtained when olivine was activated with KOH. The samples treated with only olivine added to the resistance with olivine grains, whereas the samples activated with KOH added resistance with both olivine grains and chemical bonds up to a rate of 521% in the case of 20% olivine being used. Morphological and mineralogical analyses were performed to analyze the chemical bonds formed after the reaction. Stabilization with olivine substitutes a sustainable method of improvement that provides benefits such as reducing carbon dioxide emissions and controlling waste.

**Keywords:** clay; KOH; olivine; soil improvement; sustainability; waste management





## 1. Introduction

One of the most common problems in geotechnical engineering is weak soils, which are often unable to bear the stresses transferred from structures [1]. However, the growth of cities and the increase in population corresponds to both a need to evaluate sites with such soils and the urgency of managing waste materials deposited into the environment [2]. Due to their high cost, the improvement of existing weak soils is preferred instead of using high-quality materials. In general, however, the high water content and low workability of these soils present challenges for construction projects. Civil infrastructure is often built on fragile soils, which must be modified to withstand the imposed loads. Therefore, various reinforcement methods have been proposed to improve the bearing capacity of fragile soils [3]. Among these methods, stabilization with additives has become very common in terms of easy application and rapidity [4]. The chemical stabilization of soil is the process of improving the technical properties of the soil. By creating interparticle chemical interactions that bind soil particles together, these methods can increase cohesion, shear strength, and structural stability [5]. Cement-stabilized soil (known as low-strength concrete) consists of aggregates bound together by a stabilizer, which is cement, to form a hard and rigid mass [6]. The selection of additive type mainly relies on soil characteristics, environmental climate, the aim of the improvement, environmental benefits, and cost-effectiveness [7]. Calcium-based conventional binders such as cement, lime, and fly ash are widely used in soil stabilization [8–12]. However, as the sulfate concentration in the environment increases,

these materials allow the formation of harmful minerals such as ettringite and thausamatite, thus negatively affecting the soil strength. The fact that groundwater contains sulfates proves that these materials may not give positive results all the time or on all soils [13,14]. In recent studies, it has also been proven that sulfate-containing soils cause excessive swelling and the deterioration of pavement balance when stabilized with cement or lime [15,16]. For this reason, it is extremely important to be careful in the application of materials with high calcium content, such as lime and cement, on sulfate-rich soils. Despite the great success of these calcium-based additives, they have been criticized not only for the negative environmental effects (carbon dioxide emissions) associated with their production but also for their cost, including high energy consumption [8]. One of the most important disadvantages of cement use is the emission of carbon dioxide ($CO_2$) into the atmosphere. Carbon dioxide, the main greenhouse gas from fossil fuels and human activities, causes climate change and global warming [17]. The utilization of these materials as additives leads to disastrous environmental impacts due to low carbon emissions and, therefore, reduces the quality of social life with environmental pollution [18]. For all these reasons, the demand for new, environmentally friendly materials to be used as additives in the stabilization of soils is increasing [19].

A new eco-friendly material called olivine has been added to the chemical additives used for stabilization purposes. Olivine, which has recently been shown as an efficient material in soil stabilization due to its high MgO content and its capacity to capture carbon dioxide, is actually a mineral formed from dunites [20]. Although this additive material, which was created as an alternative material due to the disadvantages of cement, was used as an aggregate in concrete before, it has recently started to be used on soils as well [17,21]. The use of the material as a stabilizer on the soil is still being discussed, and it has been seen that the studies mainly aim at the stabilization of high-plasticity clays [19,21,22].

Emmanuel et al. [8] investigated the interaction between landfill leachate and olivine-treated marine clay to evaluate their chemical compatibility in geotechnical applications. In order to find the optimum water contents, the samples were prepared using three different compression energies (reduced, standard, and modified compaction), and soil–olivine mixtures were prepared with the determined optimum water contents. In the study, the olivine ratio was increased up to 35% and experiments such as volumetric shrinkage stress (VSS), unconfined compressive strength (UCS), and hydraulic conductivity (HC) were carried out. According to the test results, the optimum olivine content was determined as 30% at three different compression ratios. In parallel with this, the unconfined compressive strength (UCS) of the samples, hydraulic conductivity (HC), and volumetric shrinkage stress (VSS) values were improved. The main reason for this improvement was explained as the formation of hydrated magnesium aluminate (MAH) and hydrated magnesium silicate (MSH). In the studies carried out, it was found that the water content and plasticity index of the soil decreased with the contribution of olivine, and the dry unit weight and shear resistance increased [10,19,21].

It is thought that olivine, which is a significant source of MgO and $SiO_2$, may be a more ideal candidate for the stabilization of soils as a result of activation with other chemical materials. In addition, one of the biggest advantages of olivine stabilization, which is used as a binder in the presence of a strong alkali (such as sodium hydroxide (NaOH) or potassium hydroxide (KOH)), is that it increases the carbonization potential [9]. Alkali activation is the process of dissolving aluminosilicate materials with alkaline activators such as potassium hydroxide (KOH) or sodium hydroxide (NaOH) [23,24]. Alkali-activated materials can have low calcium (F-class fly ash) or high calcium (C-class fly ash) or ground granulated blast furnace slag (GGBS) content [25,26].

In a study, olivine ($Mg_2SiO_4$) was activated with potassium hydroxide (KOH) in order to improve the natural soil; samples with a KOH-activated olivine mixture (5–20% by weight, 10 M KOH) were kept at different curing times [11]. The results of the study showed that the shear strength of high-plasticity clay (CH) treated with olivine in the

presence of KOH was 120 kPa in 7 days from 960 kPa and increased up to 7400 kPa in 90 days.

In another study conducted by Fasihnikoutalab et al. [9] the effects of carbon dioxide pressure and alkaline concentration (5–20% by weight, 10 M NaOH) between 7 and 90 days were investigated. With the use of olivine in the presence of NaOH, the shear strength of high-plasticity clay aged for 90 days increased up to 40%.

In previous studies, high-plasticity soils treated with olivine, which offers a more sustainable improvement than other non-environmental materials against the hazards induced by the low strength of weak soil, were tested under two different conditions, both with and without activation. However, due to the insufficient number of studies, it has been determined that the subject has been mostly examined for clays with high plasticity and there are not enough studies for clays with low plasticity. Therefore, the present study aimed to determine the undrained shear strength of a low-plasticity clay under the influence of different confining pressures in terms of representation of field conditions. In addition, in order to accelerate the curing time, an activator material, potassium hydroxide (KOH), was used. In this context, we aimed to determine the effective mixing ratios and curing time for the stabilization of the undrained shear strength of the soil type used. To detail the obtained chemical bonds between soil and additive and related compositions, Field Emission Scanning Electron Microscopy and Energy Dispersive Spectrum tests were conducted.

## 2. Materials and Method

### 2.1. Materials

#### 2.1.1. Soil

In this study, a low-plasticity clayey soil taken from Kocaeli (Turkey) was studied. The sample taken to be used in the experiments was passed through a 4.75 mm sieve (No.4). Liquid limit, plastic limit, and pycnometer experiments were carried out with the part passing through the No. 40 sieve. The sample containing 4% sand and 96% fine-grain material was classified as low-plasticity clay (CL) according to the Unified Soil Classification System [27]. The physical properties of the soil are shown in Table 1.

**Table 1.** Physical properties of the soil.

| Feature | Unit | Value | Standard |
|---|---|---|---|
| Fine Content (FC) | % | 96 | BS 410 [28] |
| Clay Fraction (CF) | % | 8 | BS 410 [28] |
| Liquid Limit (LL) | % | 41 | ASTM D 4318 [27] |
| Plastic Limit (PL) | % | 17 | ASTM D 4318 [27] |
| Plasticity Index (PI) | % | 24 | ASTM D 4318 [27] |
| Optimum Water Content (OMC) | % | 17.5 | ASTM D 467 [29] |
| Density | $g/cm^3$ | 2.7 | ASTM D 854 [30] |
| Max. Dry Unit Weight (MDD) | $kN/m^3$ | 17.8 | ASTM D 4647 [29] |

#### 2.1.2. Olivine

Olivine material was procured as a waste material from the Yıldırım Holding factory located in Elazığ, Turkey. Olivine is a silicate mineral with the composition $(Mg,Fe)_2SiO_4$. It comes from dunite, one of the first differentiated igneous rocks to crystallize from mantle melts. The use of this material for soil improvement, together with waste disposal, provides environmental benefits and strengthens sustainability as well as prevents soil-induced potential risks by increasing shear strength. The chemical properties and specific gravity determined by Energy Dispersive Spectrum (EDS) analyses are given in Table 2. Olivine, which had large particle sizes when supplied, was reduced to smaller grain sizes before being added to the soil in order to increase the chemical surface reactivity between it and the soil. After the size reduction process was applied without allowing a chemical change, the materials passing through the 212 μm sieve were used as an additive material. Figure 1

also shows the grain size distribution curve of olivine and the curve of natural soil before and after screening.

**Table 2.** Olivine's chemical and physical properties.

| Feature | Unit | Value |
|---|---|---|
| Mg | % | 35.86 |
| Si | % | 24.58 |
| O | % | 35.77 |
| Fe | % | 3.79 |
| Specific Gravity (Gs) | — | 3.30 |

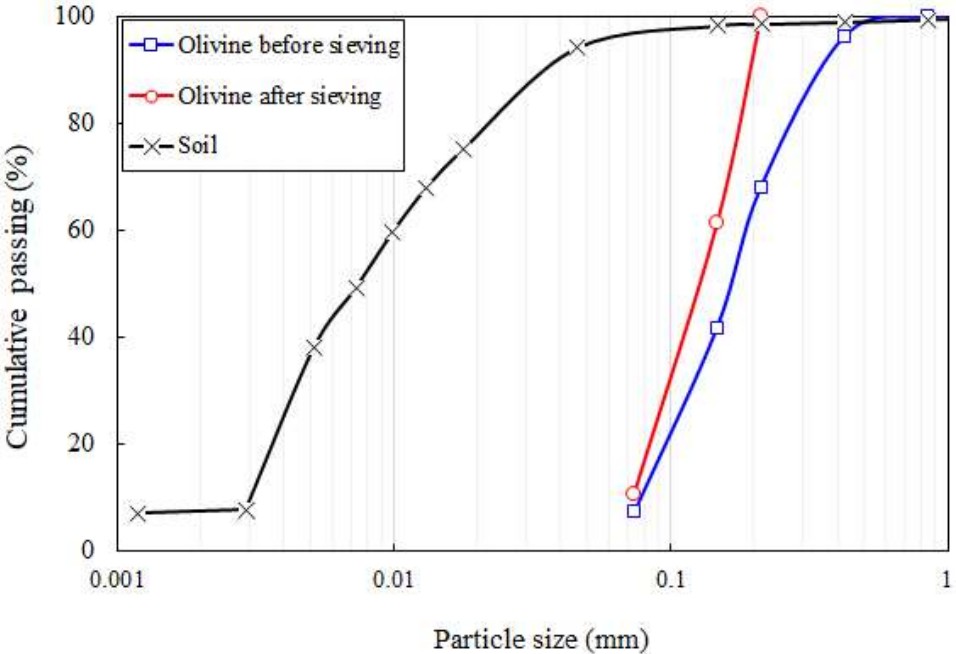

**Figure 1.** Natural soil and olivine grain distribution curves.

### 2.1.3. Potassium Hydroxide

In this study, potassium hydroxide (KOH) was used as the activator due to its well-known activity. Potassium hydroxide is a small, granular alkaline compound of white to slightly yellow color. This compound, obtained from the electrolysis of potassium chloride, creates an exothermic reaction when mixed with water or polar solvents. It is used depending on the water temperature and concentration [31]. When polypropylene fiber and KOH are added to a silty soil, the soil swelling potential and optimum water content decrease, while the maximum dry density and undrained shear resistance increase by changing the mineral structure due to the presence of K ions [32]. Although sodium hydroxide (NaOH) is a cheaper alkali activator than potassium hydroxide, it was determined that potassium hydroxide, which was determined to be more effective in strength, was preferred more frequently in concrete and soils. Potassium hydroxide-treated soil can provide a significant increase in strength in the long run [33]. Considering this, it was decided that the olivine material should be activated with this compound. The materials used in the study are given in Figure 2.

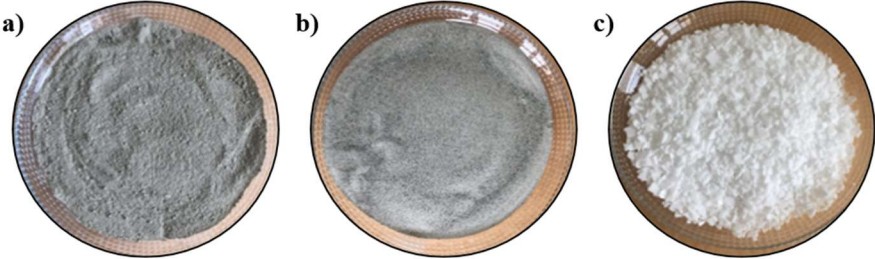

**Figure 2.** Ingredients: (**a**) soil, (**b**) olivine, (**c**) potassium hydroxide (KOH).

*2.2. Method*

2.2.1. Classification Experiments

In this study, preliminary experiments—sieve analyses, hydrometer, pycnometer, plastic limit, and liquid limit (Casagrande and fall-cone method)—were performed to classify the soil. The results and standards are shown in Table 1.

2.2.2. Compaction Test

Compaction is the process of laying the soil in layers and reducing the voids in it by compacting it. In this study, all samples were compacted using Harvard miniature compaction. In order to determine the optimum water content and maximum dry unit weight for each sample, 600 gr samples were taken, and the samples were mixed with different proportions of water (5%, 10%, 15%, and 20%) until they became homogeneous. Afterward, the mixture was placed in the mold equally in four layers. During the compression, as a result of the calculations based on the volume of the mold, corresponding to the standard Proctor energy, a 1.51 kg rammer was dropped freely from a height of 310 mm, and 10 blows were applied to each layer. For the fourth layer, the last 10 blows were completed by attaching the collar in order for the compression to be effective. After a total of 40 blows were completed, the necessary measurements of the compacted sample were taken. In order to determine the optimum water content and maximum dry unit weight, the same procedure was repeated for the mixtures with olivine after applying the same procedures for different water contents.

2.2.3. Composition of Mixtures and Preparation of Samples

The samples in this study were categorized as natural clay, clay mixed with olivine, and clay mixed with olivine + KOH. Mixtures marked "NS" represent natural soil samples, mixes marked "S-OL" represent soil and olivine-treated samples only, and those marked "S-OL-KOH" represent olivine and potassium hydroxide (KOH)-treated samples. As a result of the experiment, samples were prepared based on natural clay and different maximum dry unit weight and optimum water content values for each mixture. The samples were mixed with 20% olivine by mass and examined in two different groups, KOH-activated and non-activated. In addition, they were kept in an airless environment for different curing periods of 1, 14, 28, and 56 days [34,35]. The KOH concentration to be used in the samples was determined as 10 molarity. The mixtures were prepared in a mold with a diameter of 3.5 cm and a height of 7 cm, taking into account the optimum water contents obtained from the compaction experiment. Due to the difference in the volume, the number of layers and blows was reduced to 1 and 6, respectively, to achieve the standard proctor energy. After the alkaline activator (KOH) was dissolved in distilled water at a concentration of 10 M, the prepared solution was allowed to cool before being added to the olivine and soil mixture. The mixtures were compacted by adding as much solution liquid as the optimum water content to the samples to be tested with activation. The samples to be tested without activation were compacted considering the optimum water content of the olivine and soil mixtures. In order to avoid any change in the initial water content, the samples were kept in a plastic storage box during the curing period and were placed in two

plastic bags beforehand. All samples were cured at room temperature during the curing period (Figure 3 and Table 3).

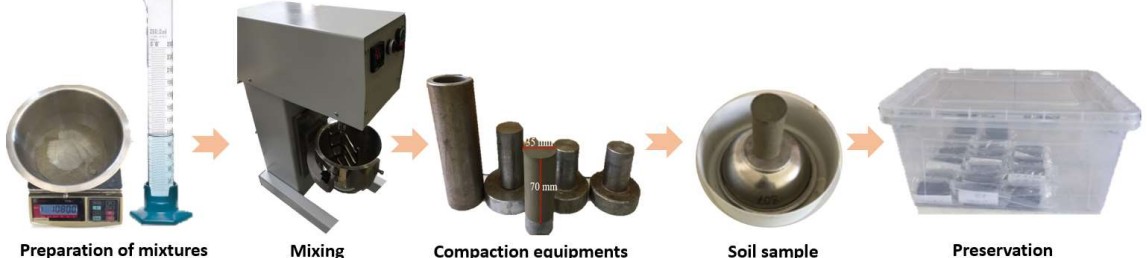

**Figure 3.** Sample preparation process.

**Table 3.** Identity of samples.

| No | Sample Symbol | Description of Samples | Soil Rate (%) | Olivine Rate (%) | KOH (mol/L) | Cure (day) |
|----|--------------|------------------------|---------------|------------------|-------------|-----------|
| 1 | NS | Natural Soil | 100 | - | - | 0 |
| 2 | S-OL-1 | Soil+ 20% Olivine | 80 | 20 | - | 1 |
| 3 | S-OL-14 | Soil+ 20% Olivine | 80 | 20 | - | 14 |
| 4 | S-OL-28 | Soil+ 20% Olivine | 80 | 20 | - | 28 |
| 5 | S-OL-25 | Soil+ 20% Olivine | 80 | 20 | - | 56 |
| 6 | S-OL-KOH-1 | Soil+ 20% Olivine+ KOH | 80 | 20 | 10 | 1 |
| 7 | S-OL-KOH-14 | Soil+ 20% Olivine+ KOH | 80 | 20 | 10 | 14 |
| 8 | S-OL-KOH-28 | Soil+ 20% Olivine+ KOH | 80 | 20 | 10 | 28 |
| 9 | S-OL-KOH-56 | Soil+ 20% Olivine+ KOH | 80 | 20 | 10 | 56 |

### 2.2.4. Unconsolidated–Undrained Triaxial compression test (UU)

A triaxial compression test (UU) was applied, in accordance with ASTM D2850-15 [36], to determine the unconsolidated undrained shear strength parameters of the soils. To determine these parameters, a sample must reach the maximum stress level it can withstand without collapsing under confining pressure. Firstly, natural soil samples were subjected to UU tests at confining pressures of 100 kPa, 200 kPa, 300 kPa, and 400 kPa and a shear rate of 0.8 mm/min. From the results of the compaction, since the 20% olivine mixture has the highest dry density, the shear strength parameters were continued on it. Then, they were kept at different curing times (1, 14, 28, and 56 days), and experiments with and without the activator for olivine and soil mixtures were carried out at the same confining pressures. In this study, the UU test was performed on all samples. Details of the test are shown in Figure 4. The load was applied according to the undrained conditions, assuming fast loading.

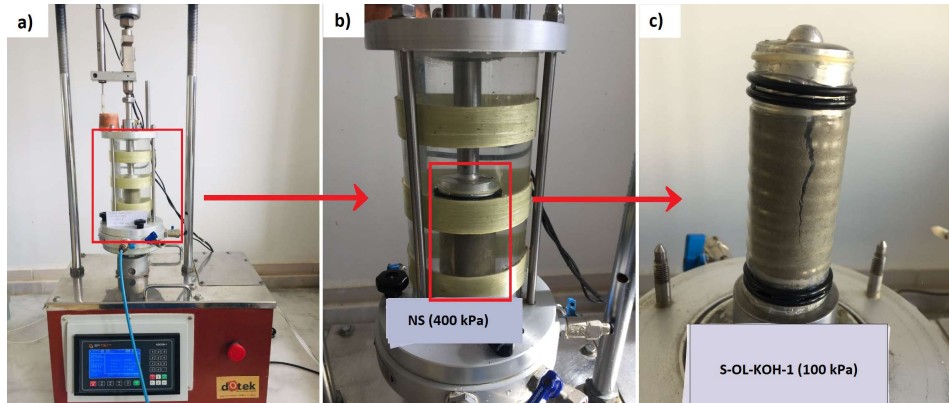

**Figure 4.** UU experiment: (**a**) test device, (**b**) test sample, (**c**) the moment of failure.

### 2.2.5. Microscopic Examinations

Microstructural analyses such as Field Emission Scanning Electron Microscopy (FE-SEM) and Energy Dispersive Spectrum (EDS) were conducted to provide evidence for the behavior at maximum strength for the natural samples, olivine-treated, and olivine + KOH-treated specimens aged in a 56-day curing period. The samples were completely dry, powdered, and solid, taken in two forms at the time of preparation and coating. The maximum sample diameter was 70 mm and the thickness was 50 mm. All analyses were performed in the Sakarya University laboratory with a detector device (Spectro Analytical Instruments, Ametek Materials Analyses Division; Model: Octane Plus, DOM: 2013).

## 3. Results and Discussions

### 3.1. Compaction Test

The compaction curves from which the maximum dry unit weights and optimum water content values of the soils examined in the study are observed are shown in Figure 4. In line with the results, it is observed that while the maximum dry unit weight (MDD) increases with increasing olivine, the optimum water content (OMC) decreases and, accordingly, the strength of the soil increases. Compared to the natural soil, the increases in the MDD value of the 10%, 15%, and 20% olivine-treated soil samples were 3.14%, 6.78%, and 9.14%, and the decrease in their OMC was 8.57%, 14.29%, and 20%, respectively. The reason for the increase in MDD can be explained by both the high specific gravity and particle sizes of olivine. The decrease in OMC is due to the low water absorption capacity of olivine compared to clay soil. These significant changes have also been observed in previous studies [19,21] (Figure 5 and Table 4).

**Table 4.** Compaction test data.

| Sample | Olivine Rate (%) | OMC (%) | MDD (kN/m³) | Reduction in OMC (%) | Increase in MDD (%) |
|--------|-----------------|---------|-------------|---------------------|---------------------|
| NS | 0 | 17.5 | 17.83 | - | - |
| S-OL-10 | 10 | 16 | 18.39 | 8.57 | 3.14 |
| S-OL-15 | 15 | 15 | 19.04 | 14.29 | 6.78 |
| S-OL-20 | 20 | 14 | 19.46 | 20 | 9.14 |

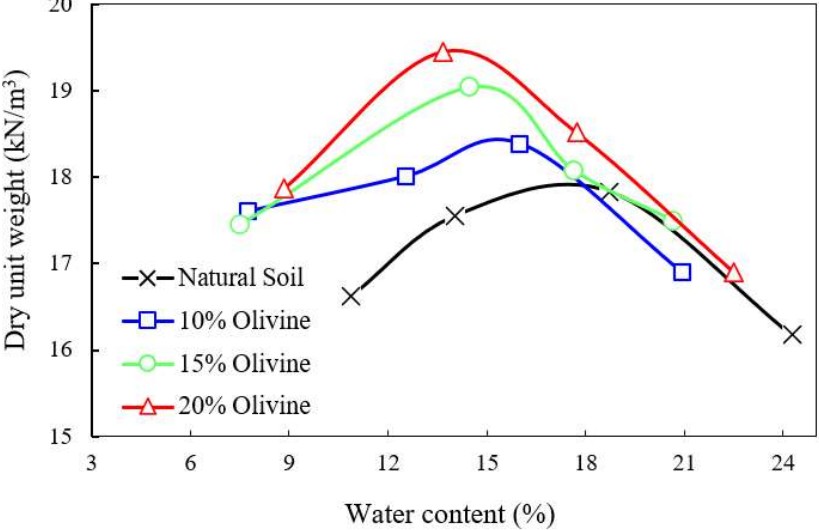

**Figure 5.** Compaction curves for olivine-free or -poor soils and soils with olivine additives (10–20%).

### 3.2. Unconsolidated–Undrained Triaxial Compression Test (UU)

The effect of olivine, potassium hydroxide (KOH), and their combination on the shear strength of clayey soil was examined by triaxial experiments (UU) at different curing days under different confining pressures. The results of the experiments are shown in Tables 5 and 6 and Figures 6–12. Natural soil (NS), olivine-treated soil (S-OL), and olivine + KOH-activated soil (S-OL-KOH) samples were subjected to triaxial compressive tests (UU) to determine the undrained shear resistance parameters.

**Table 5.** Max. deviator stresses for S-OL and S-OL-KOH.

| | Cell Pressure (kPa) | | | |
|---|---|---|---|---|
| **Sample** | **100** | **200** | **300** | **400** |
| | **Maximum Deviator Stresses** | | | |
| NS | 169 | 179 | 184 | 173 |
| S-OL-1 | 246 | 251 | 271 | 349 |
| S-OL-14 | 292 | 327 | 339 | 356 |
| S-OL-28 | 327 | 321 | 317 | 373 |
| S-OL-56 | 365 | 371 | 370 | 383 |
| S-OL-KOH-1 | 287 | 331 | 358 | 404 |
| S-OL-KOH-14 | 403 | 448 | 482 | 517 |
| S-OL-KOH-28 | 645 | 775 | 877 | 851 |
| S-OL-KOH-56 | 1050 | 1226 | 1258 | 1321 |

**Table 6.** Max. deviator stresses increase rate and shear resistance parameters.

| | | Increase in Deviator Stress (%) Cell Pressure (kPa) | | | | Shear Resistance Parameters | |
|---|---|---|---|---|---|---|---|
| **Sample** | **Days** | **100** | **200** | **300** | **400** | $c_u$**(kPa)** | $\phi_u$**(°)** |
| NS | 0 | - | - | - | - | 88 | 0 |
| S-OL-1 | 1 | 45.56 | 40.22 | 47.28 | 101.73 | 100 | 6.34 |
| S-OL-14 | 14 | 72.78 | 82.68 | 84.24 | 105.78 | 122 | 6.20 |
| S-OL-28 | 28 | 93.49 | 79.33 | 72.28 | 115.61 | 130 | 5.40 |
| S-OL-56 | 56 | 115.98 | 107.26 | 101.09 | 121.39 | 175 | 1.59 |
| S-OL-KOH-1 | 1 | 69.82 | 84.92 | 94.57 | 133.53 | 110 | 8.53 |
| S-OL-KOH-14 | 14 | 138.46 | 150.28 | 161.96 | 198.84 | 150 | 10.20 |
| S-OL-KOH-28 | 28 | 281.66 | 332.96 | 376.63 | 391.91 | 200 | 17.57 |
| S-OL-KOH-56 | 56 | 521.30 | 584.92 | 583.70 | 663.58 | 340 | 19.63 |

In Figure 6, two different behaviors (ductile and brittle) are observed. Natural soil (NS) and olivine-treated soil (S-OL) show a ductile behavior, while olivine + KOH-activated soil (S-OL-KOH) samples have a brittle behavior. These two behavior-like situations have been observed in previous studies [19,37]. While the strains in NS samples are between 17.14% and 20%, it is determined that the strains for S-OL samples vary between 18.57% and 20% at 1 day, between 17.14% and 18.57% at 14 days, and between 15.71% and 20% at 28 and 56 days. The unit strains of S-OL-KOH samples varied between 6.28% and 8.1% at 1 day, between 5.71% and 7.14% at 14 days, and between 2.14% and 4.12% at 28 and 56 days. The deviator stress values of NS samples are lower than those of S-OL samples. The reason for this is the inclusion of olivine grains, which causes an increase in dry density, and the deviator stress values of S-OL-KOH samples are higher than the deviator stress values of S-OL samples. This is because the presence of KOH cements the clay grains by transforming the olivine into another material.

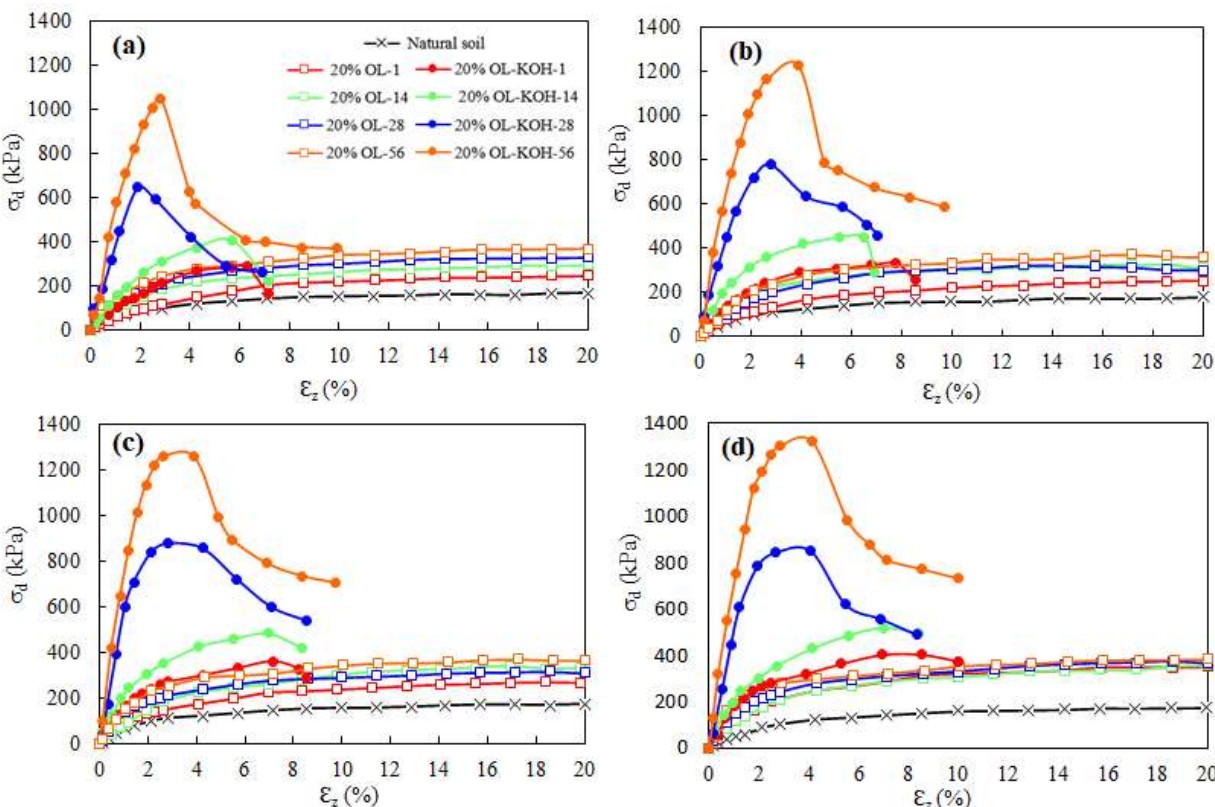

**Figure 6.** UU values of all curing days: (**a**) $\sigma_3$ = 100 kPa, (**b**) $\sigma_3$ = 200 kPa, (**c**) $\sigma_3$ = 300 kPa, (**d**) $\sigma_3$ = 400 kPa.

In Figures 7 and 8, it can be observed that the strengths of S-OL samples did not increase with time and only added to the resistance with olivine grains. It has been determined that the samples activated with KOH (S-OL-KOH) participate in the resistance both with olivine grains and chemical bonds. It was observed that the strength of S-OL-KOH samples increased with time due to chemical reactions, both in cohesion and internal friction angle, and the unit strains decreased. In this context, KOH turns olivine into a very effective binder, which increases its reactivity over time. This high strength is due to the dissolution of the olivine with the addition of KOH [11]. The strength of S-OL samples, on the other hand, remained at almost constant levels over time compared to S-OL-KOH samples. Since the samples are not saturated, the confining pressure has a slight effect on the NS samples, while it is clearly observed in the S-OL and S-OL-KOH samples. As shown in Table 6, confining pressure with reference to 100 kPa, for S-OL samples, the max. rates of increase in deviator stresses are 45.6, 72.8, 93.5, and 116.0% at 1, 14, 28, and 56 days, respectively. For S-OL-KOH samples, the max. rates of increase in deviator stresses at 1, 14, 28, and 56 days were 69.8, 138.5, 281.7, and 521.3, respectively.

From Figures 9–12, both the cohesion ($c_u$) and the internal friction angle ($\phi_u$) increased when activated with KOH, where the contribution of olivine alone was not sufficient over time. In the case where olivine was activated with KOH, a rapid increase in deviatoric stresses was observed with curing. The reason for the response from the increase could be the dissolution of Si and Al in the soil under the effect of KOH. In a previous study, a similar situation was observed when fly ash was activated with sodium hydroxide (NaOH) [38]. It turned out that at 56 days, the cohesion in the natural state ranged from 88 kPa to 340 kPa and the internal friction angle was 0° to 19°. In this study, in terms of shear resistance, the undrained internal friction angle ($\phi_u$) was added as an additional important resistance in the long run. In an unconsolidated undrained triaxial compression test, it is not advisable to give the internal friction angle for fine soils. This explains the differences in the maximum deviatoric stresses coming from the effect of increasing cell pressures. This case is the same for the natural soil in this study. But in the case of fine soils reinforced with an additive,

this may not always be the encountered situation. Here, the increase in the maximum deviatoric stresses does not come only from confinement but from the grains of the olivine and the effect of KOH, which cause a cementation of the grains. In this case, there will not only be an increase in the cohesion but in the internal friction angle as well. This explains why a friction angle appears in failure envelopes.

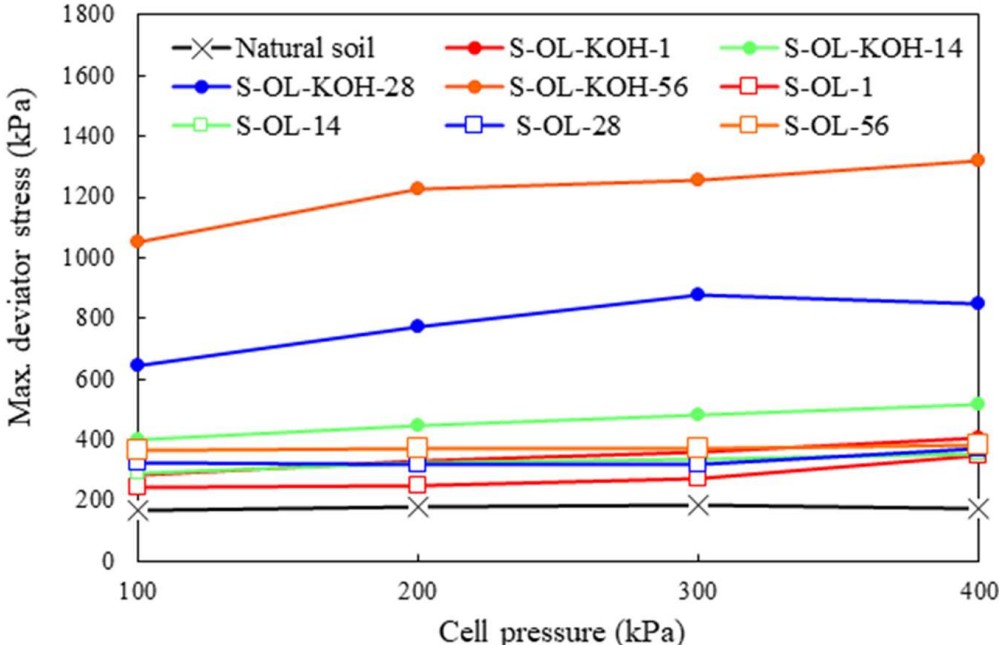

**Figure 7.** Max. deviator stresses.

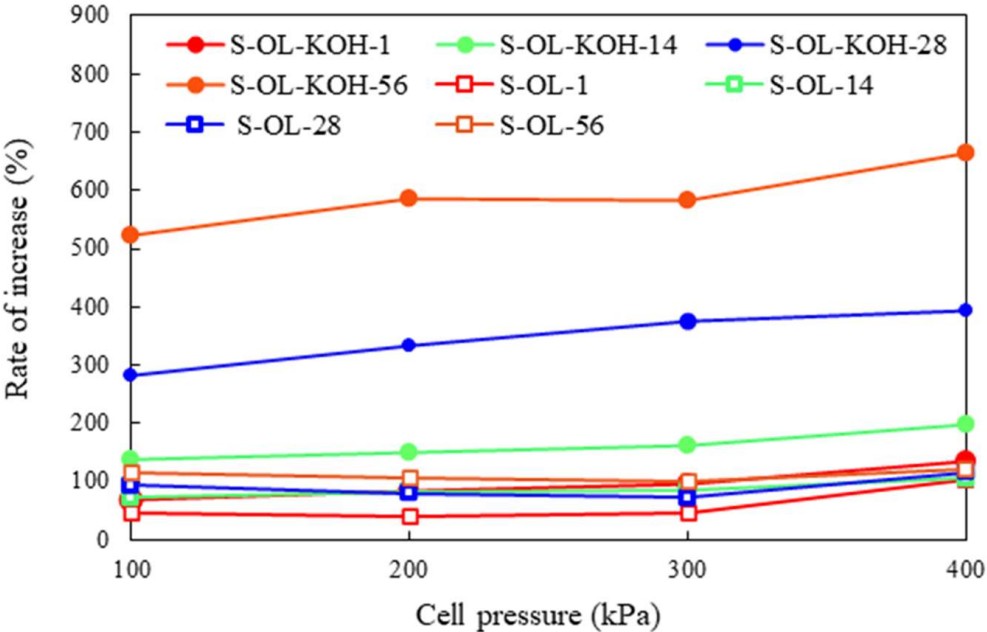

**Figure 8.** Max. deviator stresses increase rate.

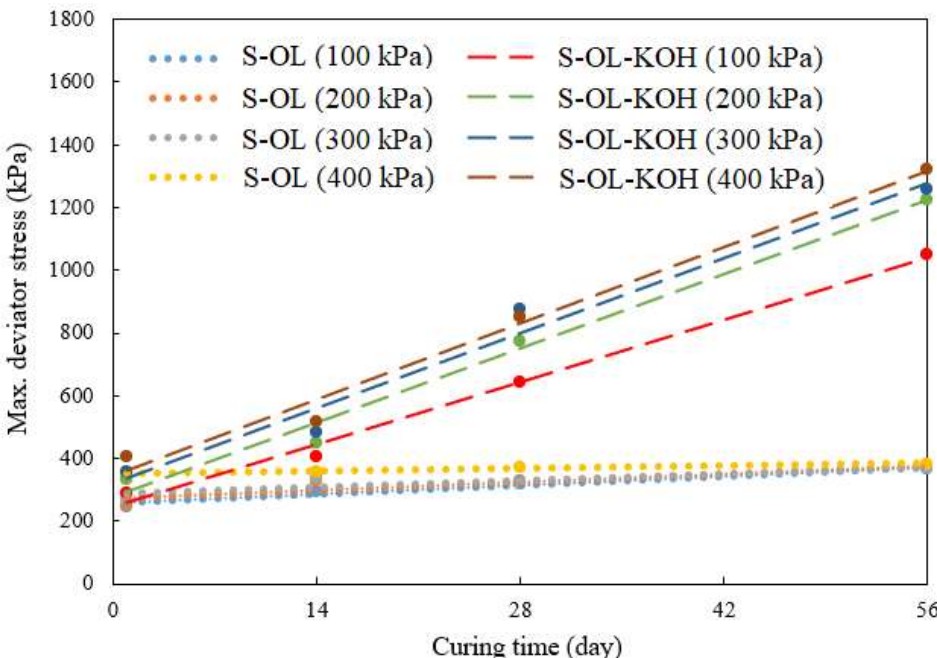

**Figure 9.** Max. deviator stresses change over time.

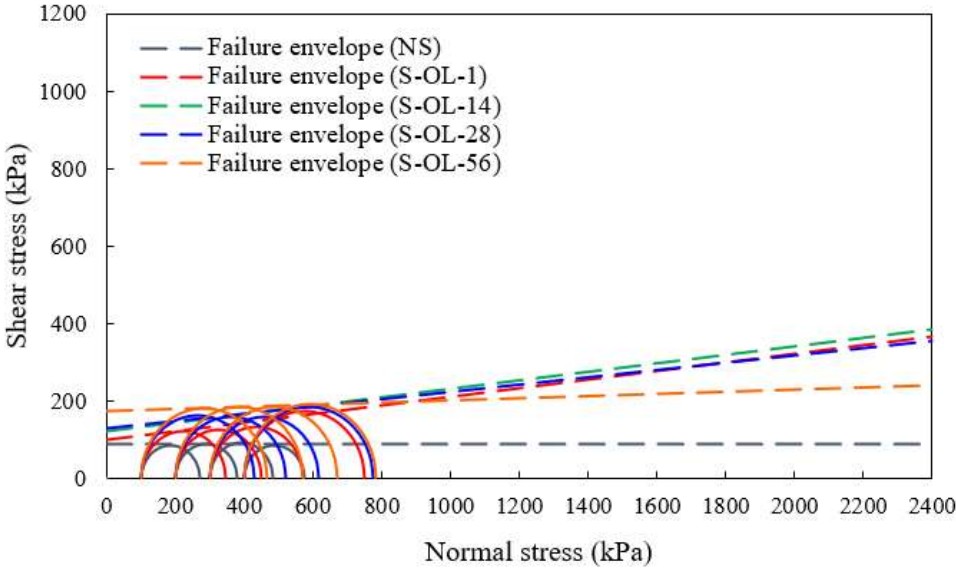

**Figure 10.** Mohr circles during all periods of curing (S-OL).

Table 7 shows the failure modes of the samples before and after the use of olivine and potassium hydroxide (KOH) under different confining pressures. The maximum deviator stress corresponds to the accepted critical axial deformation, which reaches 20% in specimens that deform uniformly without showing the failure of their cylindrical shapes with the increase in diameter and decrease in height (NS and S-OL) in the specimens. The behavior of these samples was observed as ductile behavior. On the other hand, specimens with stress–strain behavior (S-OL-KOH) preserved their cylindrical shape with conspicuous failure lines and show high strength with few deformations, varying according to curing times, and the behavior of these specimens is brittle [37]. In Figure 13, the visual representation of the whole tested samples is given.

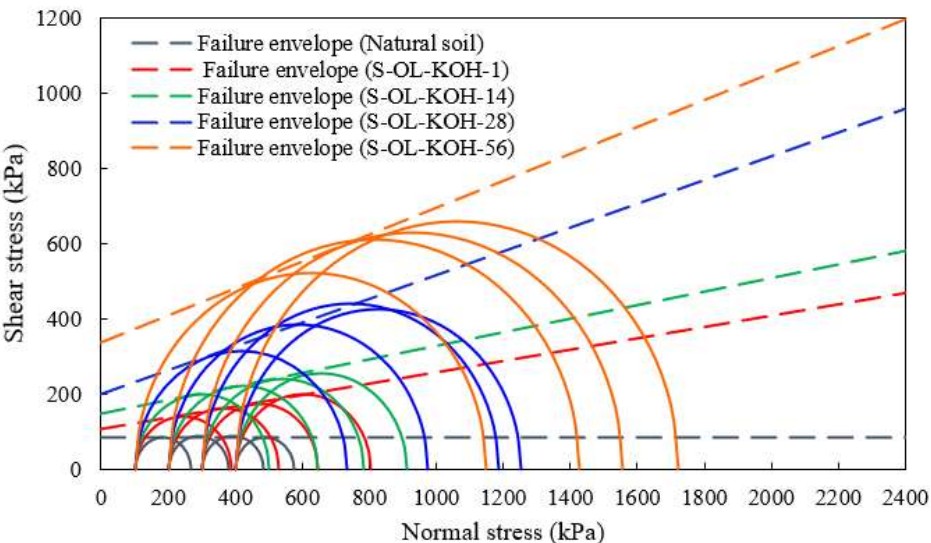

**Figure 11.** Mohr circles during all periods of curing (S-OL-KOH).

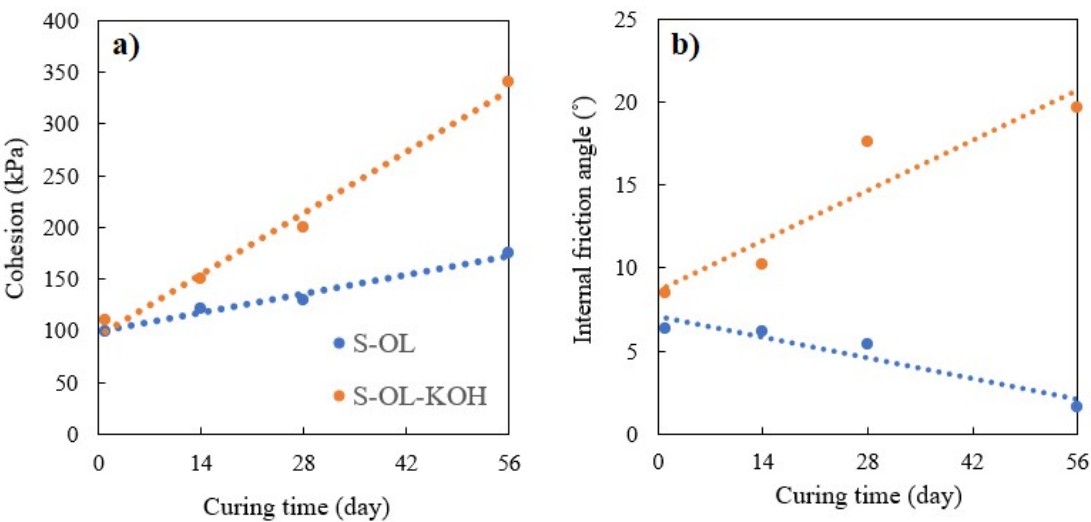

**Figure 12.** (**a**) Cohesion values over time, (**b**) internal friction angle values over time.

**Table 7.** Display of natural samples and failure modes at 56 days.

| Sample | 100 kPa | 200 kPa | 300 kPa | 400 kPa |
|--------|---------|---------|---------|---------|
| NS |  |  |  |  |

**Table 7.** *Cont.*

| Sample | 100 kPa | 200 kPa | 300 kPa | 400 kPa |
|---|---|---|---|---|
| S-OL-56 | | | | |
| S-OL-KOH-56 | | | | |

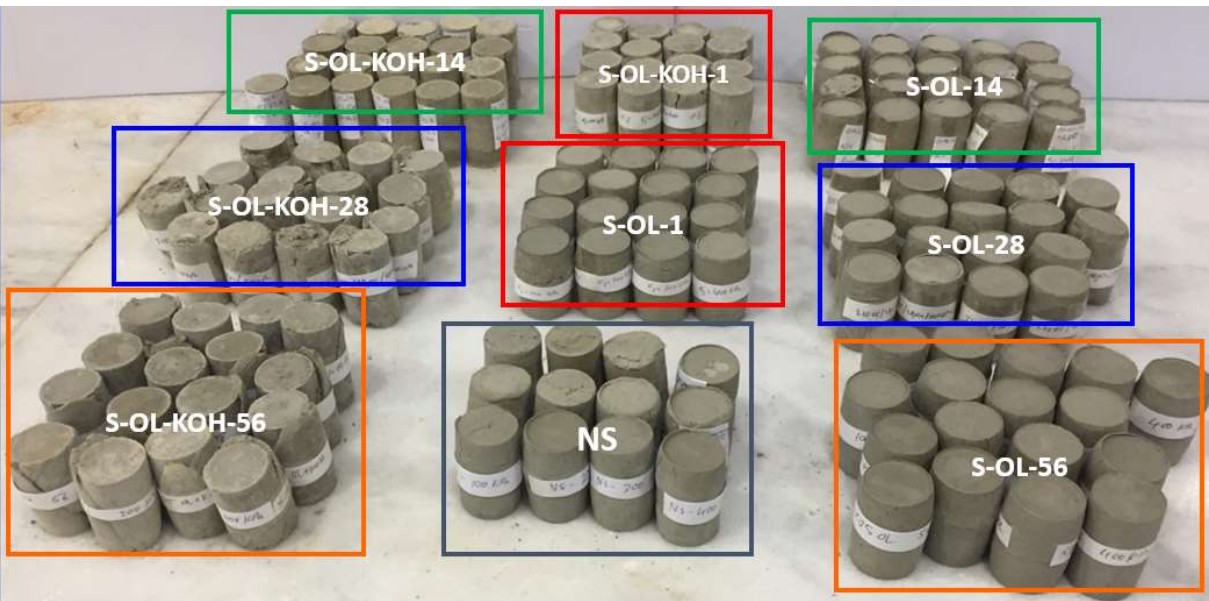

**Figure 13.** Collective sets of investigated samples subjected to mechanical tests.

*3.3. FE-SEM Analyses*

Micrographic images of natural soil (NS), 20% olivine-treated clay (S-OL), and olivine + KOH-activated samples (S-OL-KOH) after 28 days of curing are shown in Figure 14. The first state of pure olivine with angular grains and varying sizes of 60 to 600 m is given in Figure 14a. It is possible that olivine increases the internal friction angle with its angular shape and, thus, it acts in the direction of increasing the resistance. The natural soil given in Figure 14b shows the presence of widely varying visible voids, discontinuities, and a loose structure. Figure 14c shows an olivine grain surrounded by clay grains. Clay treated with olivine contains fewer voids and discontinuities compared to natural soil; however, it appears to have a denser structure. In Figure 14d, the voids and discontinuities are almost the same in olivine-treated and KOH-activated clay compared to olivine-treated clay alone. The most important difference here is that chemical bonds are formed in the presence of KOH. Images of the S-OL and S-OL-KOH samples show a compact morphology without large voids, consistent with the observed mechanical properties. As a matter of fact, in the S-OL-KOH sample, the MgO in the olivine will react chemically with the KOH, leading

to the production of magnesium hydroxide $Mg(OH)_2$ [11]. Magnesium hydroxide is the main component responsible for improving the mechanical properties of the treated soil. In previous studies, it has been observed that the formation of $Mg(OH)_2$ fills the existing pore space [39–41].

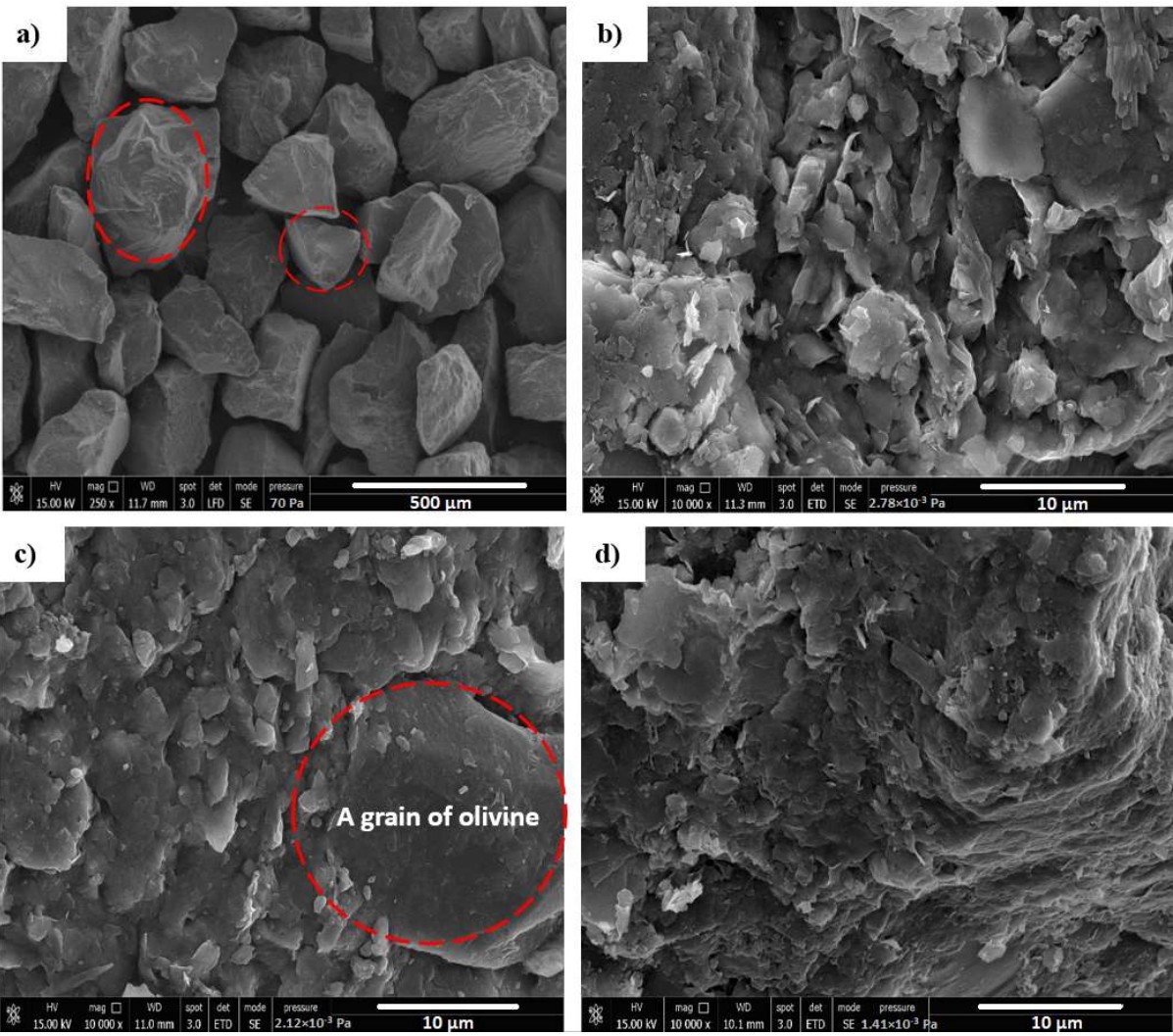

**Figure 14.** SEM analyses: (**a**) olivine, (**b**) NS sample (**c**) S-OL sample, (**d**) S-OL-KOH sample.

### 3.4. EDS Analyses

In this study, EDS analyses were performed to see the percentages of the main chemical elements. As a result of the EDS analyses, the presence of major chemical elements such as K, Si, Mg, and Al, which may cause the formation of chemical bonds, was observed. Si/Al, K/Al, and Mg/Al molar mass ratios of analyzed samples are shown in Table 8, and chemical element peaks are shown in Figure 15. The high magnesium (Mg) concentration in olivine confirms its Mg-based nature and thus facilitates hydration reactions. In addition, relatively high concentrations of Mg and K are present in the treated clay. Since the Si/Al ratio remained almost constant, it was determined that this improvement was not included in the main factors that improved the mechanical properties. When K/Al and Mg/Al values were compared with previous studies [42,43], significant increases were observed.

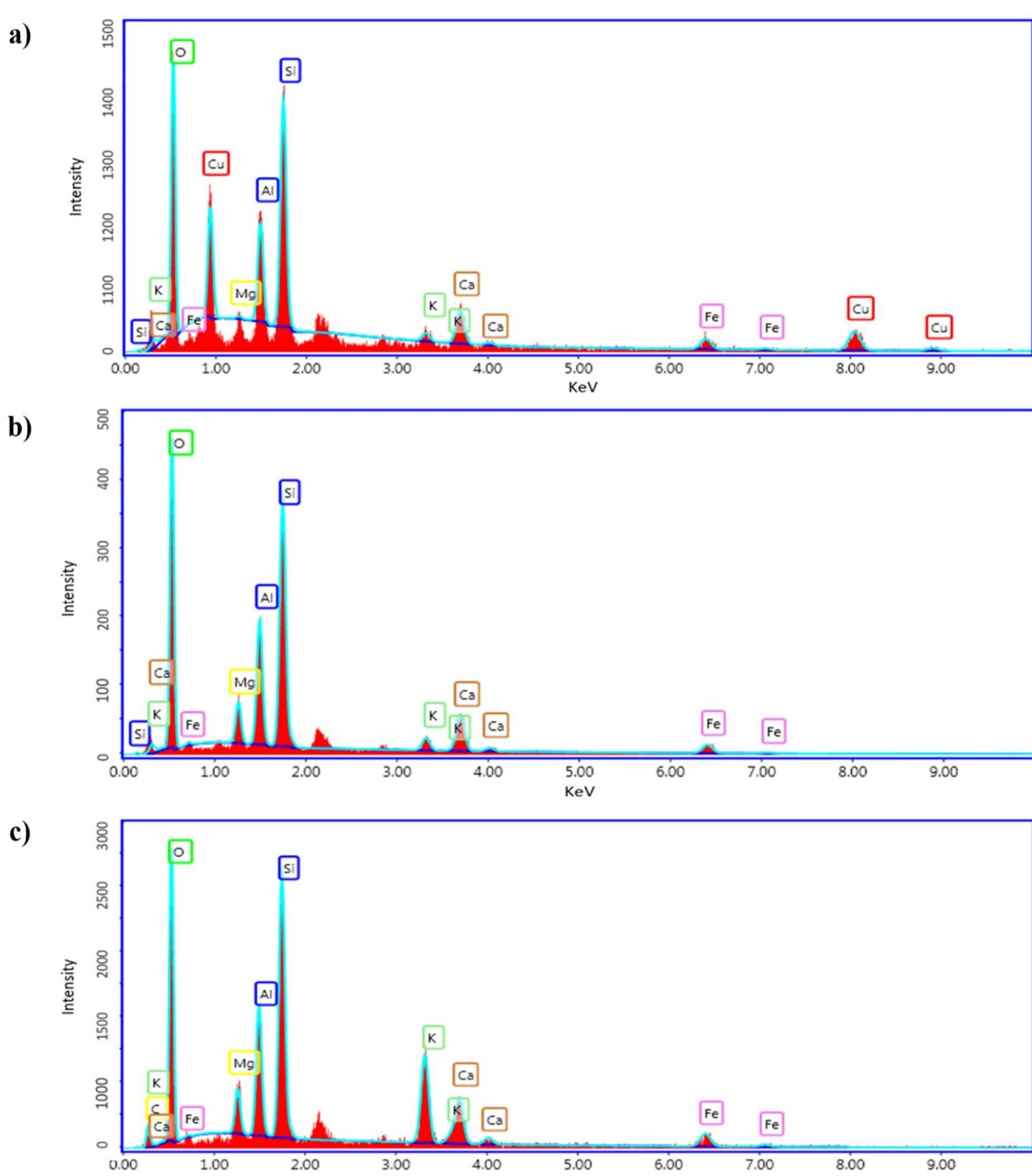

**Figure 15.** EDS analysis results: (**a**) NS sample, (**b**) S-OL sample, (**c**) S-OL-KOH sample.

**Table 8.** EDS analysis results from the samples.

| Sample | Curing Time (Day) | O (wt%) | Si (wt%) | K (wt%) | Mg (wt%) | Al (wt%) | Si/Al | K/Al | Mg/Al |
|---|---|---|---|---|---|---|---|---|---|
| NS | 0 | 31.89 | 17.84 | 1.28 | 0.44 | 8.70 | 2.05 | 0.15 | 0.05 |
| S-OL | 56 | 47.61 | 21.30 | 2.22 | 3.52 | 10.31 | 2.07 | 0.22 | 0.34 |
| S-OL-KOH | 56 | 41.83 | 16.31 | 11.53 | 2.91 | 7.82 | 2.09 | 1.47 | 0.37 |

## 4. Conclusions

In this study, an experimental study was conducted to investigate the undrained behavior of olivine-treated and olivine + KOH-activated clay under two different conditions. The main results obtained in this experimental study are listed as follows;

- As the olivine ratio increases, the maximum dry unit weight increases and the optimum water content decreases and, accordingly, the strength of the soil increases.
- In the stress–strain curve under confining stress, it was observed that the samples activated with potassium hydroxide (KOH) showed brittle behavior, while the samples that were not activated with potassium hydroxide (KOH) had a ductile behavior.
- It was observed that the strength of the olivine-treated samples (S-OL) did not increase over time compared to the potassium hydroxide (KOH)-activated samples (S-OL-KOH), and they only added to the resistance with the olivine grains. The samples activated with potassium hydroxide (S-OL-KOH) participate in the resistance both with the grains of olivine and with their chemical bonds.
- Over time, the strength of the S-OL-KOH specimens increased while the deformations decreased. However, the strength and deformation of the S-OL specimens remained constant.
- The EDX results confirm that the dissolution mechanism of olivine and the effect of the Mg/Al and K/Al ratios of the active alkali system led to an increase in strength.
- In the FE-SEM images, clay treated with olivine contained fewer voids and discontinuities compared to natural soil; however, it appeared to have a more compact structure.

As a main and essential conclusion of the paper, olivine + KOH is an energy-efficient alternative for stabilizing clayey soils, increasing shear resistance, and reducing carbon dioxide emissions, thus providing a two-way gain by providing a significant environmental benefit. The increased strength is a result of the improvement of high-plasticity clay with this material, which gives the lowest shear resistance in an undrained condition, helps the mitigation of geotechnical hazards, and at the same time eliminates waste. While the increase in the strength obtained by improving the soils is an important point in the prevention of failures originating from the soils, stabilization with olivine strengthens the work scientifically in terms of the reuse of natural resources and waste management. Therefore, this sustainable solution should have a separate scope in the literature and further studies should be performed on different soils.

**Author Contributions:** Conceptualization, A.M.T. and S.S.; Methodology, A.M.T. and S.S.; Formal analysis, A.M.T. and S.S.; Investigation, A.M.T.; Data curation, S.S.; Writing—original draft, A.M.T.; Supervision, S.S.; Project administration, S.S. All authors have read and agreed to the published version of the manuscript.

**Funding:** This research received no external funding.

**Institutional Review Board Statement:** Not applicable.

**Informed Consent Statement:** Not applicable.

**Data Availability Statement:** The datasets generated during and/or analysed during the current study are available from the corresponding author on reasonable request.

**Conflicts of Interest:** The authors declare no conflict of interest.

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
