# Peer review of "Effect of Olivine Additive on the Shear Resistance of Fine-Grained Soils: A Sustainable Approach for Risk Mitigation and Environmental Impact Reduction"

_sustainability, doi:10.3390/su151310683_

Round 1
Reviewer 1 Report
The article is well organized Therefore, I think that the article can be accepted after some minor revision. Some minor corrections given below may be applied before its acceptance:
1. Please provide short background research at the beginning of the abstract.
2. Check this (….providing a twofold) sentence in the abstract
3. Provide results value in the abstract.
4. Strokes>>>blow……. the “strokes” word can be changed to “blow”
5. Should (OL) marked for olivine.
6. It is not clearly understand what you mean from the previous studies on line 286.
7. Same for all the results; please do not only describe the results obtained. You must explain why the value changes and what the reason is for the value changing. Support your opinion with previous reputable research or publications.
8. It should be discussed how significant it is to give the internal friction angle in an undrained unconsolidated triaxial pressure test.
9. ASTM Standard should be written for the Unconsolidated - Undrained Triaxial compression test (UU).
Author Response
The authors would like to thank reviewer for her/his valuable comments. All comments were addressed in the manuscript as follows in the attached word file.

Reviewer 2 Report
Title and abstract are ok but more care should be paid for the abstract. Please re-write and make it more informative and show for the international reader what the scope, results in brief and the scientific significance in terms of improvement using olivine additives. Also, you forgot to include a sentence about waste management in the abstract.
Line 6, authors’ affiliations: It is not recommended to include your academic degree here and please write "Department" in full.
Line 7, authors’ affiliations: Sure you are proud with your language and we all are the same, but when you publish in an international journal, which uses English as the official language, please use "Turkey" instead.
Lines 57-59: You need to add that olivine is a nesosilicate mineral with the composition (Mg,Fe)2SiO4. You can separate this sentence into too and mention that olivine is common in dunite, which is an early differentiated igneous rocks crystallized from mantle melts.
Line 88: Here you suppose that olivine is forsterite without any fayalite component as solid solution. Normally, simple cationic substitution for Mg2+ by Fe2+ is common in olivine.
Line 120 in Table 1, you can used density instead of specific gravity and give the unit as g/cm3.
Lines 125-126, materials: It is true that you can determine chemical composition with EDS but how can you determine specific gravity using this technique, which is no doubt strange.
Line 126, and all over the text, please use “analyses” instead of “analyses”.
Line 152: In Figure 2, why doesn't olivine look green in the middle photo. Maybe you need to use a close view instead to obtain the actual colour of olivine.
Line 197, Figure 3: You can remove this figure because in a research paper published in an international scientific journal, you do not need to use basic things like tools used for sample preparation and curing.
Line 208: Please consider re-numbering in case Fig. 3 will be removed.
Line 221: It is much better, and highly recommended, if you separate results from discussion. First, show your results and the measurements of soil mechanical behaviour and microscopic microstructures. Then add a separate discussion section to interpret the obtained results.
Line 235: Caption of Figure 5 should be modified as follows: Compaction curves for olivine-free or -poor soils and soils with olivine additives (10-20 %).
Line 236, Table 3: This is wrong and it is not rate for sure. It is simply the amount by volume as to the engineering geology experiments you conduct for your research.
Line 274, Table 4: Some modifications are needed as shown in the attached annotated pdf.
Line 290: caption of Figure 10 should be modified as follows: Mohr circles during all periods of curing. Please do the same for the following figures.
Line 309, Fig. 13: Collective sets of investigated samples subjected to mechanical tests.
Line 330: Scale of SEM images in Figure 14 is unreadable for all so please improve.
Lines 371-373, conclusions: Try to rephrase this and split into two sentences paying more attention to waste management. Of course you give evidence for the improvement of soil shear by the use of olivine additives in different amounts and this lies actually in the scope of sustainability and better use of natural resources and their preservation. Please stress on this and give more scientific importance for your findings.
Reference list and citations in the text are OK.

Moderate polishing of English is needed.
Author Response

(The authors gave the same response as above.)

Reviewer 3 Report
This paper aimed to improve the shear resistance of clays with a sustainable material namely olivine. The undrained shear strength of the samples prepared at different curing times was tried to be determined at various confining pressures and tests were carried out in two groups, olivine-added samples only and samples using potassium hydroxide (KOH) to activate the olivine. The paper is well structured and well written. However, some comments need to be solved. It is recommended that minor revisions to address these concern.
1. Suggest adding sample preparation tables for more intuitive comparison and analysis of different samples.
2. Suggest adding pictures of the sample preparation process.
3. Figure 5: it should be 10%, 15%, and 20%.
4. This study is very meaningful, but the reference literature is relatively old. It is recommended to consider referring to Construction and Building Materials, 2022, 318:126016. (DOI:10.1016/j.conbuildmat.2021.126016).
5. And it is also suggested that the authors may consider studying the permeability characteristics of improved soil in this paper in the future, some literatures can help, such as Journal of Hydrology, 2023, 618, 129230; Journal of Hydrology, 2022,614, 128583. These papers related to infiltration (saturated or unsaturated hydraulic conductivity), pore structure, and porosity.

Author Response

(The authors gave the same response as above.)

Round 2
Reviewer 2 Report
Thank you for the improvement you made to improve the quality of your work in which the idea of adding olivine to soil to improve its shear is highly needed for sutainable applications everywhere all over the world.
Good luck in your future works.
Just need fine English polishing (very minor)